# A Full Loading-Based MVDR Beamforming Method by Backward Correction of the Steering Vector and Reconstruction of the Covariance Matrix

**Jing Zhou and Changchun Bao ***

Speech and Audio Signal Processing Laboratory, Faculty of Information Technology, Beijing University of Technology, Beijing 100124, China

* Correspondence: baochch@bjut.edu.cn; Tel.: +86-10-6739-1635

**Abstract:** In order to improve the performance of the diagonal loading-based minimum variance distortionless response (MVDR) beamformer, a full loading-based MVDR beamforming method is proposed in this paper. Different from the conventional diagonal loading methods, the proposed method combines the backward correction of the steering vector of the target source and the reconstruction of the covariance matrix. Firstly, based on the linear combination, an appropriate full loading matrix was constructed to correct the steering vector of the target source backward. Secondly, based on the spatial sparsity of the sound sources, an appropriate loading matrix was constructed to further suppress interferences. Thirdly, the spatial response power was utilized to derive a more accurate direction of arrival (DOA) of the target source, which is helpful for obtaining a more accurate steering vector of the target source and a more effective covariance matrix iteratively. The simulation results show that the proposed method can effectively suppress interferences and noise.

**Keywords:** speech enhancement; beamforming; minimum variance distortionless response (MVDR); backward calibration; interference suppression

## 1. Introduction

The minimum variance distortionless response (MVDR) beamformer has proved to be effective for suppressing interferences and noise by using an effective covariance matrix of the interference-plus-noise (CMIN) and an accurate steering vector of the target source [1–5]. Since the CMIN and the steering vector of the target source are unknown in the conventional MVDR beamformer (CMB), their applications are greatly limited. Thus, many improved MVDR beamforming methods [6–9] have been proposed successively. In the typical examples, the diagonal loading-based MVDR beamformers [9–16] have been paid more attention to recently. The purpose of the diagonal loading operation is to reduce the diffusion of the eigenvalues of the noise by loading the values on the diagonal elements of the covariance matrix of the observed signal (CMOS), which can reduce the error of the CMOS and the susceptibility to mismatch of the steering vector [9,10]. However, a larger loading value lets the CMB degrade to a general fixed beamformer; on the contrary, a smaller loading value can hardly improve the performance of the CMB [11,12].

In order to find the appropriate loading value, many diagonal loading methods have been proposed in recent decades. They can be divided into three categories. The first one is the single loading parameter-based diagonal loading method, whereby the loading matrix is the product of the loading value and unit matrix and loaded on the CMOS, such as the Hoerl–Kennard–Baldwin (HKB) method [9] and the load-to-white-noise ratio (LNR) method [10]. Although these two methods can reduce the diffusion of the eigenvalues of the noise with the appropriate loading value, they are susceptible to specific parameters. Hence, some specific parameter-free methods have been proposed, such as the spatially matched filter (SMF) method [11] and the bounded perturbation regularization (BPR) method [12].

Nevertheless, these methods usually depend on a priori assumptions or constraints, which can easily lead the loading value to be too large or too small. The second category of diagonal loading methods is the multiple loading parameters-based diagonal loading methods, which is usually a linear combination of the CMOS and the loading matrix, such as the general linear combination (GLC) method [13] and the noise reduction preprocessing into a truncated minimum mean square error (NRP-TMMSE) method [14]. Although this category of methods has a higher flexibility for designing the loading value, they are also sensitive to the number of microphones and snapshots. Moreover, the optimal objectives of these methods do not focus on making the loaded covariance matrix approach to the CMIN, which limits the improvement on the self-cancellation of the main lobe. The third category is the iterative diagonal loading methods, which tries to improve the performance of the CMB by performing diagonal loading iteratively, such as the parameter-free Landweber iteration (PFLI) method [15] and the iterative diagonally loaded sample matrix inverse (IDL-SMI) method [16]. This category of methods utilizes iteration to compensate for the limitation of a single diagonal loading operation but is sensitive to the loading value or the weighting value used for the iterative update.

Currently, the available diagonal loading-based MVDR beamforming methods are suitable for cases in which the steering vector of the target source does not mismatch or only has a small mismatch [17]. However, the studies in recent years have still mainly been focused on designing the loading value [18–21], which is intent on obtaining an effective compromise between the noise reduction and interference suppression, whereas the essential problem of the mismatch of the steering vector is ignored. In addition, the diagonal loading-based methods also ignore the influence of the off-diagonal elements.

Based on the aforementioned problems, a full loading-based MVDR beamforming method is proposed in this paper, which combines the advantages of multiple loading parameters and iteration. In this method, a full and an appropriate loading matrix was constructed to correct the steering vector of the target source backward and further suppress the interferences, respectively. In addition, spatial response power was used to derive a more accurate direction of arrival (DOA) of the target source.

The remaining parts of this paper are organized as follows: Section 2 describes the conventional diagonal loading methods; Section 3 gives the proposed full loading method; Section 4 shows the simulations and the results analysis; and Section 5 draws the conclusions.

## 2. Conventional Diagonal Loading Methods

Considering a free far-field case including one target source, $G$ interference sources and environmental noise, a uniform linear array (ULA) with $M$ ($G < M$) microphones was used to pick up the multi-channel speech signals. Assuming that the target source, interference sources and noise are mutually independent, by minimizing the output power of the interference plus noise with the constraint that the target source is distortionless, the MVDR beamformer at the $k$th frequency bin of the $l$th frame was built as follows [1–5]:

$$\min_{\mathbf{w}} \mathbf{w}^H \mathbf{R}_{\text{IN}} \mathbf{w} \quad \text{s.t.} \ \mathbf{w}^H \mathbf{a}_0 = 1 \tag{1}$$

where $\mathbf{w} \in \mathbb{C}^{M \times 1}$ is the$\in$ weighting vector of the MVDR beamformer, $\mathbf{R}_{\text{IN}} \in \mathbb{C}^{M \times M}$ is the real CMIN, $\mathbf{a}_0 \in \mathbb{C}^{M \times 1}$ is the real steering vector of the target source, the superscript "$H$" indicates the conjugate transpose and the symbol "$\mathbb{C}$" indicates the complex space.

Equation (1) is commonly solved by the Lagrange multiplier method [3,5] as follows:

$$\mathbf{w} = \frac{\mathbf{R}_{\text{IN}}^{-1} \mathbf{a}_0}{\mathbf{a}_0^H \mathbf{R}_{\text{IN}}^{-1} \mathbf{a}_0} \tag{2}$$



Since $\mathbf{R}_{\text{IN}}$ is unknown in practice, it is usually replaced by the CMOS $\mathbf{R}_{\text{XX}} \in \mathbb{C}^{M \times M}$, i.e.:

$$\mathbf{R}_{\text{IN}} \rightarrow \mathbf{R}_{\text{XX}} = \frac{1}{J} \sum_{j=l+1-J}^{l} \mathbf{x}(j)\mathbf{x}^H(j) \tag{3}$$

where $J$ is the number of the snapshots and $\mathbf{x}(j) \in \mathbb{C}^{M \times 1}$ is the $j$th snapshot of the observed signal.

Substituting Equation (3) into Equation (2), the weighting vector of the CMB can be expressed as follows by using the estimated steering vector $\hat{\mathbf{a}}_0 \in \mathbb{C}^{M \times 1}$ of the target source:

$$\mathbf{w}_{\text{CMB}} = \frac{\mathbf{R}_{\text{XX}}^{-1}\hat{\mathbf{a}}_0}{\hat{\mathbf{a}}_0^H \mathbf{R}_{\text{XX}}^{-1}\hat{\mathbf{a}}_0} \tag{4}$$

The purpose of diagonal loading methods is to improve the performance of the CMB by loading values on the diagonal elements of the $\mathbf{R}_{\text{XX}}$ before the inverse operation [8,9,11], so that the general form of the weighting vector $\mathbf{w}_{\text{DL}} \in \mathbb{C}^{M \times 1}$ of the MVDR beamformer realized by the single loading parameter-based diagonal loading method can be expressed as follows:

$$\mathbf{w}_{\text{DL}} = \frac{(\mathbf{R}_{\text{XX}} + \xi\mathbf{I})^{-1}\hat{\mathbf{a}}_0}{\hat{\mathbf{a}}_0^H(\mathbf{R}_{\text{XX}} + \xi\mathbf{I})^{-1}\hat{\mathbf{a}}_0} \tag{5}$$

where $\xi$ is the loading value and $\mathbf{I} \in \mathbb{C}^{M \times M}$ is a unit matrix.

Obviously, $\xi$ is the key parameter for improving the CMB. When $\xi = 0$, Equation (5) degenerates to the CMB. When $\xi$ is much larger than the powers of the sound sources, Equation (5) degenerates to the delay and sum (DAS) beamformer [9–12]. Table 1 gives six diagonal loading methods of the CMB including HKB, LNR, SMF, BPR, GLC and NRP-TMMSE, respectively, where the symbol "$||\ ||_2$" indicates the 2-norm operation.

**Table 1.** Six Diagonal Loading Methods of the CMB.

| Methods | Diagonal Loading | Methods | Diagonal Loading |
|---------|------------------|---------|------------------|
| HKB | $\dfrac{(M-1)(\mathbf{A}\hat{\boldsymbol{\eta}}-\mathbf{b})^2}{\|\hat{\boldsymbol{\eta}}\|_2^2}$ | LNR | $10^{\frac{\zeta_{\text{LNR}}}{10}}\sigma_{\mathbf{w}}^2$ |
| SMF | $\bar{\mathbf{a}}_0^H\mathbf{R}_{\text{XX}}\bar{\mathbf{a}}_0$ | BPR | $\dfrac{\delta\|\mathbf{A}\hat{\boldsymbol{\eta}}-\mathbf{b}\|_2}{\|\hat{\boldsymbol{\eta}}\|_2}$ |
| GLC | $\min\limits_{\alpha,\beta}\left\{\|\alpha\mathbf{R}_{\text{XX}}+\beta\mathbf{I}-\mathbf{R}_{\text{EXP}}\|_2^2\right\}$ | NRP-TMMSE | $\min\limits_{\alpha,\beta}\left\{\|\alpha\mathbf{R}_{\text{XX}}+\beta\mathbf{I}-\mathbf{R}_{\text{T}}\|_2^2\right\}$ |

From Table 1, we can see that the HKB uses the difference between the output $\mathbf{b} \in \mathbb{C}^{M \times 1}$ of the fixed filter and the output $\mathbf{A}\hat{\boldsymbol{\eta}}$ of the blocking filter to solve the loading values [9], where $\mathbf{A} \in \mathbb{C}^{M \times (M-1)}$ and $\hat{\boldsymbol{\eta}} \in \mathbb{C}^{(M-1) \times 1}$ are the outputs of the blocking matrix and weighting vector, respectively. The performance of the HKB is greatly affected by the number of microphones.

The LNR designs the loading value by the power $\sigma_{\text{W}}^2$ of white noise and a specific parameter $\zeta_{\text{LNR}}$ (usually set to 10) [10], which aims to reduce the divergence of the eigenvalues that correspond to noise and make a compromise between the noise reduction and interference suppression by loading a value related to the power of white noise. However, the LNR is sensitive to the estimated power of white noise and the specific parameter $\zeta_{\text{LNR}}$.

The SMF uses the output power ($\bar{\mathbf{a}}_0^H\mathbf{R}_{\text{XX}}\bar{\mathbf{a}}_0$, where $\bar{\mathbf{a}}_0 \in \mathbb{C}^{M \times 1}$ is the normalized steering vector of the target source) of the spatially matched filter as the loading value [11]. Since the output power of the SMF may contain the power of the target source, the SMF may cause the loading value to be too large. Therefore, the SMF-based MVDR beamformer is easy to degenerate into a fixed beamformer.

Based on the HKB, the BPR introduces a perturbation matrix and a constraint factor to avoid the influence of the number of microphones [12]. However, the approximate solution of the constraint factor $\delta$ is likely to be too small, which limits the improvement of performance on the CMB.

The GLC uses the non-negative shrinkage parameters $\alpha$ and $\beta$ to combine the CMOS $\mathbf{R}_{XX}$ and the unit matrix $\mathbf{I}$ linearly, where these two parameters are solved by minimizing the error between the loaded covariance matrix and the covariance matrix $\mathbf{R}_{EXP} \in \mathbb{C}^{M \times M}$ that we expect [13]. Obviously, the GLC has higher flexibility for designing the loading value, but it is also sensitive to the number of microphones and snapshots.

Meanwhile, the NRP-TMMSE uses the denoised CMOS $\mathbf{R}_T \in \mathbb{C}^{M \times M}$ and convex optimization technique to solve the loaded covariance matrix and a more accurate steering vector of the target source [14]. It is an optimized method of the GLC. However, the optimization objectives of the GLC and NRP-TMMSE do not focus on making the loaded covariance matrix approach the CMIN, which limits the improvement of the problem of the self-cancellation of the main lobe.

Moreover, the diagonal loading iteration of the PFLI [15] is built as follows:

$$\begin{cases} \mathbf{w}_{LI}^{\langle 0 \rangle} = 0 \\ \mathbf{w}_{LI}^{\langle i_{\text{iteration}} \rangle} = \mathbf{w}_{LI}^{\langle i_{\text{iteration}} - 1 \rangle} + \alpha_{LI} \left( \hat{\mathbf{a}}_0 - \mathbf{R}_{XX} \mathbf{w}_{LI}^{\langle i_{\text{iteration}} - 1 \rangle} \right) \end{cases} \tag{6}$$

where $\mathbf{w}_{LI} \in \mathbb{C}^{M \times 1}$ is the weighting vector of the beamformer, $\mathbf{0}$ is a zero vector, $<i_{\text{iteration}}>$ is the index of the iteration and $\alpha_{LI} \{0 < \alpha_{LI} < (||\mathbf{R}_{XX}||_2)^{-1}\}$ is a relaxation factor. Obviously, the effectiveness of the PFLI depends on the $\hat{\mathbf{a}}_0$, the selection of $\alpha_{LI}$ and the number of iterations.

Furthermore, the IDL-SMI [16] is based on the assumption that the diagonal loading operation can improve the CMOS, so the backward correction of the steering vector of the target source is built as follows:

$$\mathbf{w}_{LD} = \frac{\mathbf{R}_{LD}^{-1} \hat{\mathbf{a}}_0}{\hat{\mathbf{a}}_0^H \mathbf{R}_{LD}^{-1} \hat{\mathbf{a}}_0} \triangleq \mathbf{w}_{CMB} = \frac{\mathbf{R}_{XX}^{-1} \tilde{\mathbf{a}}_0}{\tilde{\mathbf{a}}_0^H \mathbf{R}_{XX}^{-1} \tilde{\mathbf{a}}_0} \tag{7}$$

The solution of the corrected steering vector of the target source $\tilde{\mathbf{a}}_0$ is given by:

$$\tilde{\mathbf{a}}_0 = \frac{\tilde{\mathbf{a}}_0^H \mathbf{R}_{XX}^{-1} \tilde{\mathbf{a}}_0}{\hat{\mathbf{a}}_0^H \mathbf{R}_{LD}^{-1} \hat{\mathbf{a}}_0} \mathbf{R}_{XX} \mathbf{R}_{LD}^{-1} \hat{\mathbf{a}}_0 = \alpha_{IDL} \mathbf{R}_{XX} \mathbf{R}_{LD}^{-1} \hat{\mathbf{a}}_0 \tag{8}$$

Here, to avoid the influence of the constant $\alpha_{IDL}$, $\tilde{\mathbf{a}}_0$ is normalized as $\bar{\mathbf{a}}_0 = \sqrt{M} \tilde{\mathbf{a}}_0 / ||\tilde{\mathbf{a}}_0||_2$.

Since the improvement of the single operation of Equation (8) is quite limited, the iterative operation is used on the IDL-SMI. Its termination conditions are as follows [16]:

$$\begin{cases} \left\| \bar{\mathbf{a}}_0^{\langle i_{\text{iteration}} \rangle} - \bar{\mathbf{a}}_0^{\langle i_{\text{iteration}} - 1 \rangle} \right\|_2 < \delta_{IDL} \\ \dfrac{\left| \left( \bar{\mathbf{a}}_0^{\langle i_{\text{iteration}} \rangle} \right)^H \hat{\mathbf{a}}_0 \right|}{\left\| \bar{\mathbf{a}}_0^{\langle i_{\text{iteration}} \rangle} \right\|_2 \|\hat{\mathbf{a}}_0\|_2} \geq \min \left( \dfrac{\left| \hat{\mathbf{a}}_{0,-\Delta}^H \hat{\mathbf{a}}_0 \right|}{\|\hat{\mathbf{a}}_{0,-\Delta}\|_2 \|\hat{\mathbf{a}}_0\|_2}, \dfrac{\left| \hat{\mathbf{a}}_{0,+\Delta}^H \hat{\mathbf{a}}_0 \right|}{\|\hat{\mathbf{a}}_{0,+\Delta}\|_2 \|\hat{\mathbf{a}}_0\|_2} \right) \end{cases} \tag{9}$$

where $\delta_{IDL}$ is the parameter of the iterative increment. $\hat{\mathbf{a}}_{0,-\Delta} \in \mathbb{C}^{M \times 1}$ and $\hat{\mathbf{a}}_{0,+\Delta} \in \mathbb{C}^{M \times 1}$ are the steering vectors related to the angles of $\theta_0 - \theta_\Delta$ and $\theta_0 + \theta_\Delta$, respectively, $\theta_0$ is the estimated DOA of the target source and $\theta_\Delta$ is a small angular interval. Although the IDL-SMI seems to be very effective in reducing the mismatch of the steering vector, when the diagonal loading method does not play an optimization role, the estimated steering vector $\bar{\mathbf{a}}_0^{<\text{end}>} \in \mathbb{C}^{M \times 1}$ may be deviated from the real steering vector of the target source $\mathbf{a}_0 \in \mathbb{C}^{M \times 1}$.

## 3. The Proposed Method

### 3.1. Framework of the Proposed Method

To improve the aforementioned problems, a full loading-based MVDR beamforming method is proposed, and its framework mainly includes three modules as shown in Figure 1.

In Module 1, an improved GLC (IGLC) with a full loading matrix is used to correct the steering vector of the target source backward so that the distortion of the target source can be improved. In Module 2, an appropriate loading matrix based on the steered response power of the uncertain sets is used to reconstruct the covariance matrix so that the interferences can be further suppressed. In Module 3, based on the broadband spatial response power of the designed MVDR beamformer, the more accurate DOA of the target source is derived; thus, the initial steering vector of the target source used in Module 1 and the uncertain sets used in Module 2 can be more accurate. By iterating the above three modules, we can obtain the finally converged DOA of the target source; thus, a robust MVDR beamformer can be obtained.

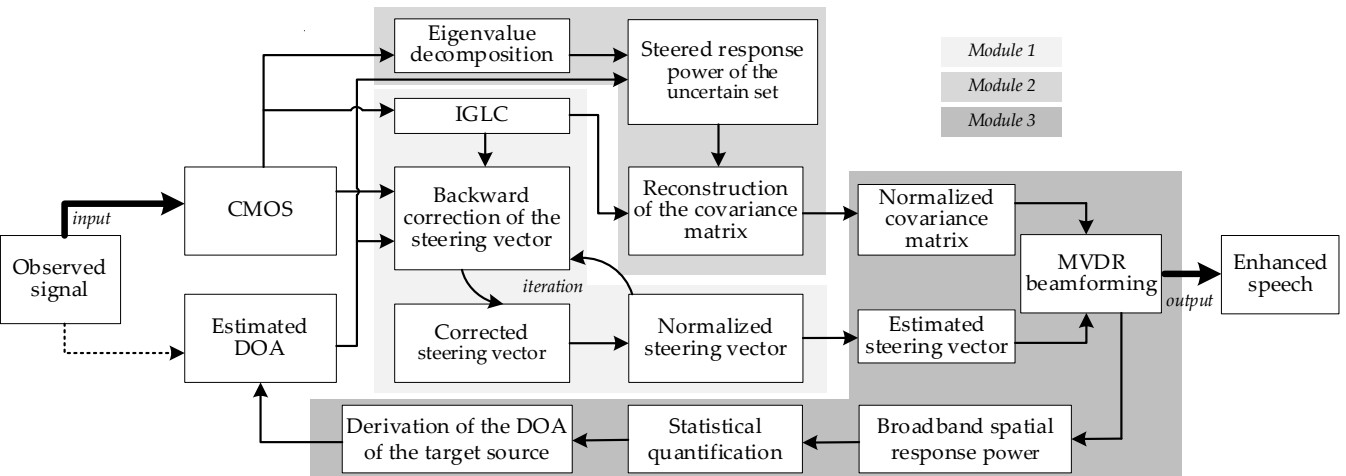

**Figure 1.** The framework of the proposed method.

### 3.2. The Improved GLC Method

#### 3.2.1. Full Loading of the Covariance Matrix

Since the GLC can improve the CMOS by minimizing the error between the loaded covariance matrix $\mathbf{R}_{\mathrm{GLC}} \in \mathbb{C}^{M \times M}$ and $\mathbf{R}_{\mathrm{EXP}}$, we chose it as the basic model. Firstly, the diagonal loading matrix was replaced by the full loading matrix, which can reduce the components of the target source and the error of the off-diagonal elements. The full loading-based covariance matrix is defined as follows:

$$
\begin{aligned}
\mathbf{R}_{\mathrm{IGLC}} &= \boldsymbol{\alpha}\mathbf{R}_{\mathrm{LM1}} + \boldsymbol{\beta}\mathbf{R}_{\mathrm{LM2}} \\
&= [\alpha_1, \alpha_2, \alpha_3][\mathbf{R}_{\mathrm{XX}}, \mathbf{R}_{\mathrm{white}}, \mathbf{I}]^T - [\beta_1, \beta_2, \cdots, \beta_P][\mathbf{R}_{\mathrm{SS},1}, \mathbf{R}_{\mathrm{SS},2}, \cdots, \mathbf{R}_{\mathrm{SS},P}]^T \\
&= \alpha_1 \tilde{\mathbf{R}}_{\mathrm{XX}} + \alpha_2 \mathbf{R}_{\mathrm{white}} + \alpha_3 \mathbf{I} - \sum_{m=1}^{P} \beta_m \mathbf{R}_{\mathrm{SS},m}
\end{aligned}
\tag{10}
$$

where $\boldsymbol{\alpha} = [\alpha_1, \alpha_2, \alpha_3]$ and $\boldsymbol{\beta} = [\beta_1, \beta_2, \ldots, \beta_P]$ are the vectors of the non-negative shrinkage parameters. $\mathbf{R}_{\mathrm{LM1}} = [\mathbf{R}_{\mathrm{XX}}, \mathbf{R}_{\mathrm{white}}, \mathbf{I}]^T$, $\mathbf{R}_{\mathrm{LM2}} = [\mathbf{R}_{\mathrm{SS},1}, \mathbf{R}_{\mathrm{SS},2}, \ldots, \mathbf{R}_{\mathrm{SS},P}]^T$. $\mathbf{R}_{\mathrm{white}} \in \mathbb{C}^{M \times M}$ is the basic covariance matrix of white noise and $\mathbf{R}_{\mathrm{SS},m} \in \mathbb{C}^{M \times M}$ is the covariance matrix corresponding to the $m$th eigenvalue of $\mathbf{R}_{\mathrm{XX}}$, which can be approximatively regarded as the covariance matrix of the $m$th source signal. The superscripted "$T$" indicates the transpose operation. $P$ is the number of the significantly large eigenvalues [22,23].

In Equation (10), $\mathbf{R}_{\mathrm{white}}$ is used to reduce the influence of the off-diagonal elements, $\mathbf{I}$ is used to realize the diagonal loading and $\mathbf{R}_{\mathrm{SS},m}$ is used to reduce the components of the target source. Among them, $\mathbf{R}_{\mathrm{white}}$ and $\mathbf{R}_{\mathrm{SS},m}$ need to be solved. Here, we use the noise subspace related to the smallest eigenvalue to estimate $\mathbf{R}_{\mathrm{white}}$. Thus, $\mathbf{R}_{\mathrm{white}}$ and $\mathbf{R}_{\mathrm{SS},m}$ can be estimated as $\mathbf{R}_{\mathrm{white}} \approx \lambda_M \mathbf{v}_M \mathbf{v}_M^H$ and $\mathbf{R}_{\mathrm{SS},m} = \lambda_m \mathbf{v}_m \mathbf{v}_m^H$, respectively. Where $\lambda_1 \geq \lambda_2 \geq, \ldots, \geq \lambda_m \geq, \ldots, \geq \lambda_M$ are the eigenvalues of $\mathbf{R}_{\mathrm{xx}}$, $\mathbf{v}_M \in \mathbb{C}^{M \times 1}$ and $\mathbf{v}_m \in \mathbb{C}^{M \times 1}$ are the

eigenvectors related to $\lambda_M$ and $\lambda_m$, respectively. Hence, the optimization of the IGLC is built as follows:

$$\min_{\alpha,\beta} E\left\{\|\mathbf{R}_{\text{IGLC}} - \mathbf{R}_{\text{IN}}\|_2^2\right\} = \min_{\alpha,\beta} E\left\{\left\|\alpha_1\mathbf{R}_{\text{XX}} + \alpha_2\mathbf{R}_{\text{white}} + \alpha_3\mathbf{I} - \sum_{m=1}^{P}\beta_m\mathbf{R}_{\text{SS},m} - \mathbf{R}_{\text{IN}}\right\|_2^2\right\}$$

(11)

However, the covariance matrix $\mathbf{R}_{\text{IN}}$ of the interference-plus-noise is unknown. Here, we use the uncertain set-based method [24,25] to estimate $\mathbf{R}_{\text{IN}}$. The uncertain set-based method uses spatial power (also called Capon power) to estimate the covariance matrix of each sound source. So, the $\mathbf{R}_{\text{IN}}$ can be estimated as follows:

$$\tilde{\mathbf{R}}_{\text{IN}} = \mathbf{R}_{\text{XX}} - \int_{\boldsymbol{\phi}_{\text{T}}} \frac{\mathbf{a}(\theta)\mathbf{a}^H(\theta)}{\mathbf{a}^H(\theta)\mathbf{R}_{\text{XX}}^{-1}\mathbf{a}(\theta)}\,d\theta$$

(12)

where $\boldsymbol{\phi}_{\text{T}}$ is the uncertain set of the target source and it can be established according to the estimation method of the initial DOA of the target source given in [24]. $\mathbf{a}(\theta)\in\mathbb{C}^{M\times 1}$ is the steering vector corresponding to angle $\theta$.

### 3.2.2. Solution of the Non-Negative Shrinkage Parameters

Although $\tilde{\mathbf{R}}_{\text{IN}}$ is not accurate enough, it can provide a good guide for the IGLC of Equation (11). Once $\tilde{\mathbf{R}}_{\text{IN}}$ is estimated, the vectors $\boldsymbol{\alpha}$ and $\boldsymbol{\beta}$ can be determined. Substituting Equation (12) into Equation (11), the optimization is rewritten as follows:

$$
\begin{aligned}
\min_{\alpha,\beta} E\left\{\|\mathbf{R}_{\text{IGLC}} - \mathbf{R}_{\text{IN}}\|_2^2\right\} &= \min_{\alpha,\beta} E\left\{\left\|\alpha_1\mathbf{R}_{\text{XX}} + \alpha_2\mathbf{R}_{\text{white}} + \alpha_3\mathbf{I} - \sum_{m=1}^{P}\beta_m\mathbf{R}_{\text{SS},m} - \mathbf{R}_{\text{IN}}\right\|_2^2\right\} \\
&= \min_{\alpha,\beta} E\left\{\left\|\alpha_2\mathbf{R}_{\text{white}} + \alpha_3\mathbf{I} - \sum_{m=1}^{P}\beta_m\mathbf{R}_{\text{SS},m} - (1-\alpha_1)\mathbf{R}_{\text{IN}} + \alpha_1(\mathbf{R}_{\text{XX}} - \mathbf{R}_{\text{IN}})\right\|_2^2\right\} \\
&\approx \min_{\alpha,\beta}\left\langle\left\|\alpha_2\mathbf{R}_{\text{white}} + \alpha_3\mathbf{I} - \sum_{m=1}^{P}\beta_m\mathbf{R}_{\text{SS},m} - (1-\alpha_1)\tilde{\mathbf{R}}_{\text{IN}}\right\|_2^2 + \alpha_1^2 E\left\{\|\mathbf{R}_{\text{XX}} - \tilde{\mathbf{R}}_{\text{IN}}\|_2^2\right\}\right\rangle
\end{aligned}
$$

(13)

and:

$$E\left\{\|\mathbf{R}_{\text{XX}} - \tilde{\mathbf{R}}_{\text{IN}}\|_2^2\right\} \approx \frac{1}{J^2}\sum_{j=l+1-J}^{J}\|\mathbf{x}(j)\|_2^4 - \frac{1}{J}\|\tilde{\mathbf{R}}_{\text{IN}}\|_2^2$$

(14)

where the elements of $\boldsymbol{\alpha}$ and $\boldsymbol{\beta}$ are non-negative. Equation (13) is a multivariate quadratic optimization problem and can easily be solved [13,26].

### 3.3. Backward Correction of the Steering Vector of the Target Source

Once $\mathbf{R}_{\text{IGLC}}$ is obtained, the backward correction of the steering vector of the target source can be realized via the IDL-SIM method. Based on Equation (7) and the weighting vector $\mathbf{w}_{\text{IGLC}}$ of the IGLC-based MVDR beamformer, the corrected steering vector of the target source can be expressed as follows:

$$\tilde{\mathbf{a}}_0 = \mu\mathbf{R}_{\text{XX}}\mathbf{R}_{\text{IGLC}}^{-1}\hat{\mathbf{a}}_0$$

(15)

where $\mu = \left(\tilde{\mathbf{a}}_0^H\mathbf{R}_{\text{XX}}^{-1}\tilde{\mathbf{a}}_0\right)/\left(\hat{\mathbf{a}}_0^H\mathbf{R}_{\text{IGLC}}^{-1}\hat{\mathbf{a}}_0\right)$.

To avoid the influence of $\mu$, $\tilde{\mathbf{a}}_0$ is normalized as:

$$\bar{\mathbf{a}}_0 = \frac{\sqrt{M}\tilde{\mathbf{a}}_0}{\|\tilde{\mathbf{a}}_0\|_2}$$

(16)

Equation (16) can then be substituted into Equation (15), i.e., by replacing $\hat{\mathbf{a}}_0$ with $\bar{\mathbf{a}}_0$, and this procedure can be repeated until the convergence conditions of Equation (9) are satisfied.

### 3.4. Reconstruction of the Covariance Matrix

Although Equation (16) can obtain a more accurate steering vector of the target source, the error between $\mathbf{R}_{\mathrm{IGLC}}$ and $\mathbf{R}_{\mathrm{IN}}$ cannot be completely eliminated; that is, the components of the target source may still exist in $\tilde{\mathbf{R}}_{\mathrm{IN}}$. Therefore, a new covariance matrix is reconstructed to reduce the sensitivity of the self-cancellation of the main lobe. Using the reconstructed covariance matrix, we hoped the components of the target source would be reduced or that the components of the interference-plus-noise would be highlighted. Based on the eigenvalue decomposition of $\mathbf{R}_{\mathrm{XX}}$, we define the weights of the target source, interference sources and noise as $\rho_{\mathrm{target}}$, $\rho_{\mathrm{interference}}$ and $\rho_{\mathrm{noise}}$, respectively, as shown in Equation (17). Moreover, based on the spatial sparsity of the sound sources, the uncertain sets of sound sources were used to reconstruct the covariance matrix. Hence, the weights of the entire angular space can be coherently expressed as (17):

$$\begin{cases} \rho_{\mathrm{target}} = \lambda_M \\ \rho_{\mathrm{interference}} = \lambda_1 \\ \rho_{\mathrm{noise}} = \frac{1}{M-P}\sum_{i=P+1}^{M}\lambda_i \end{cases} \Rightarrow \rho(\vartheta) = \begin{cases} \rho_{\mathrm{target}}, \vartheta \in \boldsymbol{\phi}_{\mathrm{T}} \\ \rho_{\mathrm{interference}}, \vartheta \in \bigcup\limits_{g=1}^{G}\boldsymbol{\phi}_{\mathrm{I}} \\ \rho_{\mathrm{noise}}, \text{otherwise} \end{cases} \tag{17}$$

where $\vartheta$ is the angle varying from $0°$ to $180°$ and $\boldsymbol{\phi}_{\mathrm{I}}$ is the uncertain set of the interference sources.

Thus, the reconstructed covariance matrix $\mathbf{R}_{\mathrm{rec}} \in \mathbb{C}^{M \times M}$ can be calculated by the steered spatial power as follows:

$$\mathbf{R}_{\mathrm{rec}} = \mathbf{R}_{\mathrm{IGLC}} + \int_{0}^{180} \rho(\vartheta)\mathbf{a}(\vartheta)\mathbf{a}^{H}(\vartheta)d\vartheta \tag{18}$$

where $\mathbf{a}(\vartheta) \in \mathbb{C}^{M \times 1}$ is the steering vector related to angle $\vartheta$.

Similarly, in order to reduce the impact caused by the modulus of $\mathbf{R}_{\mathrm{rec}}$, Equation (18) is also normalized as follows:

$$\overline{\mathbf{R}}_{\mathrm{rec}} = \frac{\lceil \mathbf{R}_{\mathrm{IGLC}} \rceil}{\lceil \mathbf{R}_{\mathrm{rec}} \rceil}\mathbf{R}_{\mathrm{rec}} \tag{19}$$

where the symbol "$\lceil \ \rceil$" indicates the determinant operation.

### 3.5. MVDR Beamforming

By solving Equations (16) and (18) simultaneously, the weighting vector of the proposed MVDR beamforming method is calculated as follows:

$$\mathbf{w}_{\mathrm{proposed}} = \frac{\mathbf{R}_{\mathrm{rec}}^{-1}\bar{\mathbf{a}}_0}{\bar{\mathbf{a}}_0^{H}\mathbf{R}_{\mathrm{rec}}^{-1}\bar{\mathbf{a}}_0} \tag{20}$$

Thus, the enhanced speech in the time-domain can be obtained by the beamforming and inverse short-time Fourier transform.

### 3.6. DOA Deduction through the Spatial Response Power and Iteration

Although the procedures from Section 3.2 to Section 3.4 can effectively improve the performance of the CMB, the effectiveness of Equations (15) and (18) is seriously affected by the initial DOA of the target source. Therefore, spatial response power is used to derive a more accurate DOA of the target source. The spatial response power at the $k$th frequency bin of the $l$th frame can be expressed as follows:

$$\psi(\vartheta) = \mathbf{w}_{\mathrm{proposed}}^{H}\mathbf{a}(\vartheta)\mathbf{a}^{H}(\vartheta)\mathbf{w}_{\mathrm{proposed}} \tag{21}$$

By statistically quantifying the maximum of the spatial response power in each frame, the more accurate DOA of the target source can be derived. In addition, since Equations (15), (18) and (21) are not optimal, the iterative operation is used to optimize the performance of the proposed method. The iterative procedure is given as follows:

- **Step 1**: Calculate the CMOS $\mathbf{R}_{\text{XX}}$ through Equation (3);
- **Step 2**: Calculate the loaded covariance matrix $\mathbf{R}_{\text{IGLC}}$ through Equation (10);
- **Step 3**: Correct the steering vector of the target source through Equation (15) and normalize it;
- **Step 4**: Reconstruct the covariance matrix $\mathbf{R}_{\text{rec}}$ through Equation (16), and normalize it;
- **Step 5**: Calculate the weighting vector $\mathbf{w}_{\text{proposed}}$ through Equation (20);
- **Step 6**: Calculate the spatial response power $\psi(\vartheta)$ through Equation (21) and derive a new DOA of the target source;
- **Step 7**: Update the DOA of the target source and return to Step 2 to repeat the procedures from Step 2 to Step 6 until the derived DOA of the target source is converged.

The mean $\gamma$ of the DOA difference between the last three iterations and the current iteration is used for judging whether the proposed algorithm is converged, i.e.,

$$\gamma = \frac{1}{4} \sum_{\kappa=\tau-3}^{\tau} \Delta \vartheta_\kappa \tag{22}$$

where $\Delta \vartheta \kappa$ is the difference in the derived DOA between the $\kappa$th iteration and the $(\kappa-1)$th iteration, and $\tau$ is the number of the iteration. If $\gamma \leq 0.1$, the proposed algorithm converges. Otherwise, the proposed algorithm does not converge.

### 4. Simulations and Analysis

*4.1. Simulation Setup*

In the simulation, the number $M$ of microphones was set to 10, the spacing element $d$ was set to 0.02 m and the acoustic speed was 340 m/s. A TIMIT corpus [27] was used to generate the observed signal through a microphone array signal simulator given in [28]. A total of 200 utterances was randomly selected for the target speech source and 200 other utterances were randomly selected for the interference speech source. The signal-to-noise ratio (SNR) of white noise was set to 0 dB, 5 dB, 10 dB, 15 dB, 20 dB and 25 dB, respectively. The signal-to-interference ratio (SIR) was set to 0 dB. The DOAs of the target source and the interference source were randomly generated from 0° to 180° with a minimum interval of 25°. The sampling rate of the signal was 8 kHz, the frame length was set to 256 samples and a 256-point fast Fourier transform (FFT) was used. The Blackman–Harris window and Hamming window, both with a 50% overlap, were used for signal analysis and signal synthesis, respectively. The length of these two windows encompassed all 256 samples.

The signal-to-interference-plus-noise ratio (SINR) [5], perceptual evaluation of speech quality (PESQ) [29], short-time objective intelligibility (STOI) [30] and speech distortion index (SDI) [2] were used as the evaluation measures. Among them, the SINR indicates the capability of the interference suppression and noise reduction, the PESQ and STOI indicate the speech quality and intelligibility and the SDI indicates the target speech distortion. Hence, the larger the SINR, PESQ and STOI, the better the performance of the algorithm. The smaller the SDI, the better the performance of the algorithm. We compared the proposed method with the CMB of Equation (4), ideal MVDR beamformer (IMB) of Equation (2), HKB, LNR, SMF, BPR, GLC, NRP-TMMSE, PFLI and IDL-SMI.

*4.2. Comparison of Spectrograms*

Figure 2 shows a comparison of the spectrograms, where the SNR was 10 dB, the number of snapshots was 100, and the real DOAs of the target source and interference source were 70° and 20°, respectively. The initial DOA of the target source was 80°, which had an error of 10°. In Figure 2, the values of the color bar represent the logarithmic amplitude of the spectrum. Figure 2 shows that the IMB effectively recovered the target speech

source and suppressed the interference source and noise. This confirmed that the MVDR beamformer can effectively enhance the target source with an accurate steering vector and an effective CMIN. Moreover, Figure 2 also shows that the target speech was seriously distorted by the CMB and BPR methods, a little distorted by the HKB and LNR methods and slightly distorted by the rest of the methods. Meanwhile, the residual components of the interference source severely remained in the SMF, GLC, NRP-TMMSE, PFLI and IDL-SMI methods. This indicates that the proposed method has a better performance than the reference methods except for the IMB.

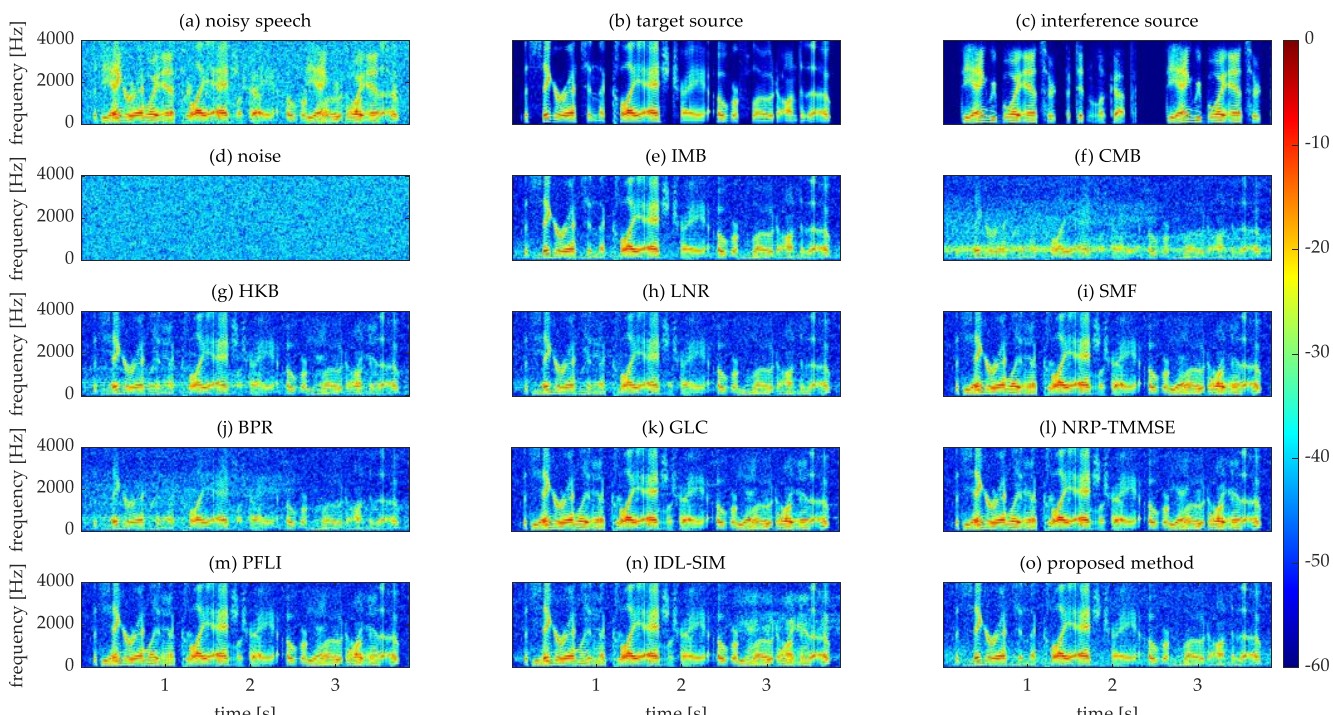

**Figure 2.** A comparison of spectrograms. The number of snapshots was 100 and the SNR was 10 dB.

### 4.3. Comparison of Beampatterns

Beampatterns were used to clarify the meaning of the spectrogram results in Section 4.2. Figure 3 shows the beampatterns and the values of the color bar represent the beampattern amplitude in dB. Figure 3 shows that the main lobes of the CMB, HKB, LNR, SMF, BPR, GLC and PFLI could not be directed at the real DOA of the target source of 70°, and the main lobe of the CMB and BPR were evidently cancelled by themselves. This explained why the target speech was seriously distorted in these two methods. Moreover, since the main lobes of the HKB, LNR, SMF, GLC and NRP-TMMSE were not directed at the real DOA of the target source, the target speech recovered by them was low-pass filtered. This explained why the target speech was slightly distorted in these five methods. Conversely, in the IMB, IDL-SIM and proposed method, the direction of the main lobe was close to the real DOA of the target source, so the target speech was well recovered.

However, the nulls of the SMF, GLC, NRP-TMMSE, PFLI and IDL-SMI were not effectively formed in the direction of the interference source, which explained why the residual components of the interference sources were maintained. Moreover, the proposed method not only made the main lobe closer to the real DOA of the target source, but also formed a null at the direction of the interference source. Hence, the proposed method has better performance for recovering the target source and suppressing the interference source.

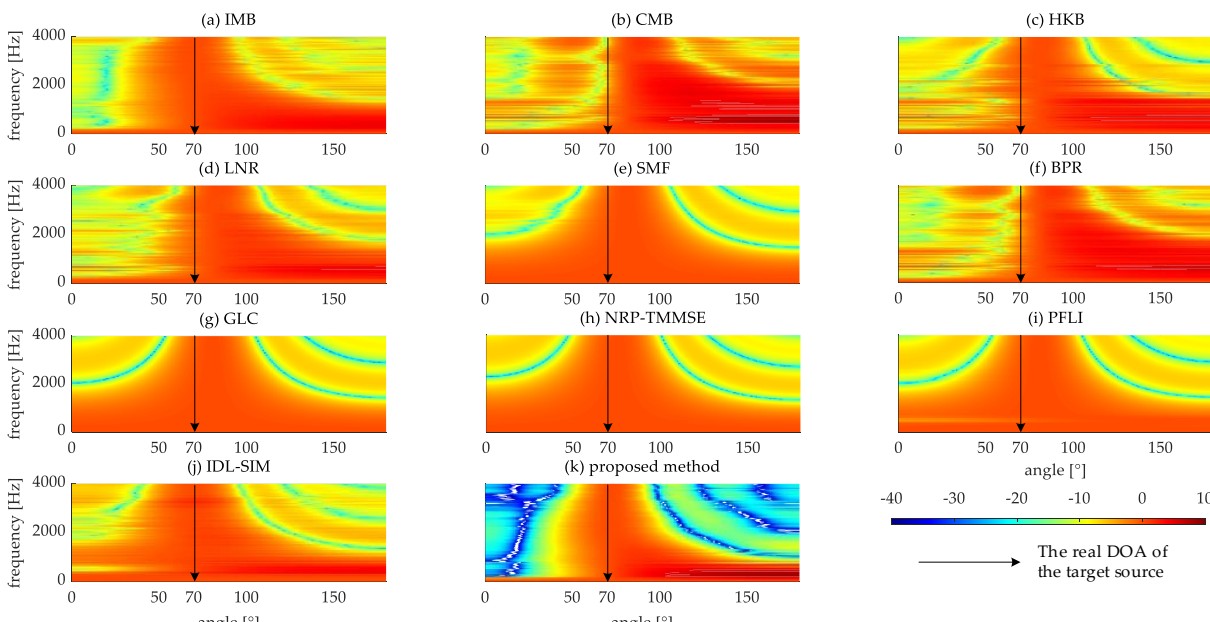

**Figure 3.** A comparison of beampatterns. The number of snapshots was 100 and the SNR was 10 dB.

### 4.4. Comparison of Evaluation Measures

In order to further evaluate the performance and robustness of the proposed method under different levels of noise, Figure 4 gives the evaluation results of the SINR, PESQ, STOI and SDI versus the input SNR (iSNR). In Figure 4, "Noisy" indicates the observed signal of the reference microphone.

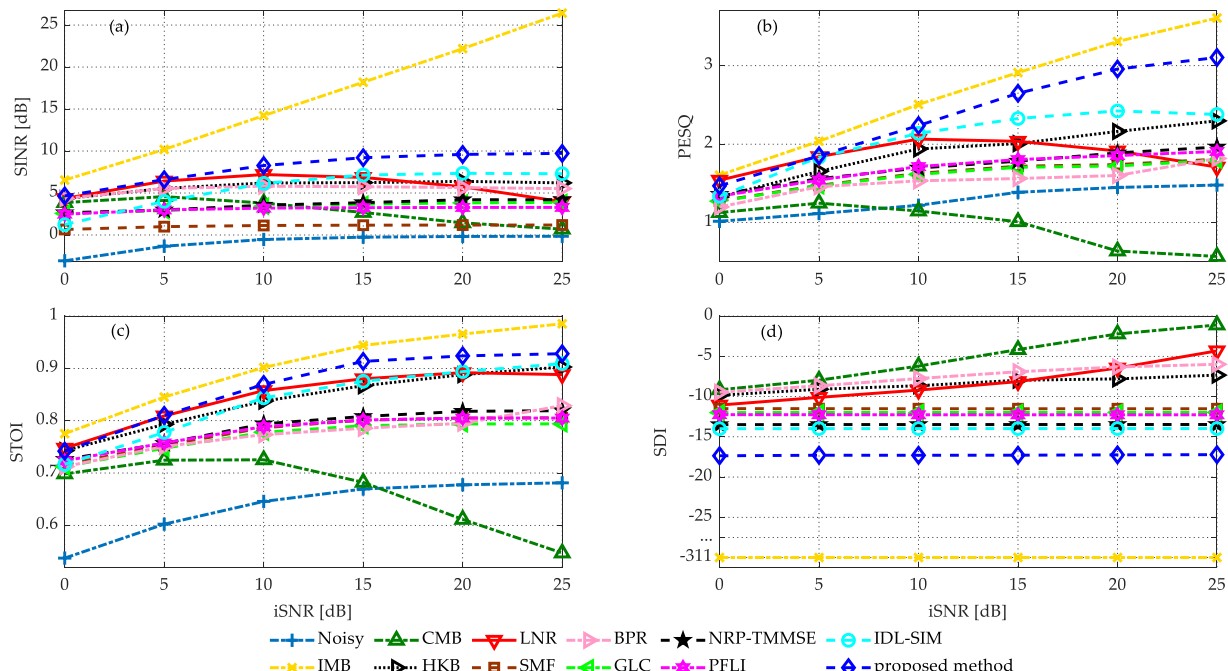

**Figure 4.** The results of the evaluation measures under different iSNRs: (**a**) SINR; (**b**) PESQ; (**c**) STOI; (**d**) SDI.

In Figure 4a–c, as the iSNR increased, the outputs of the SINR, PESQ and STOI of the IMB, HKB, SMF, BPR, GLC, IDL-SIM, NRP-TMMSE, PFLI, IDL-SIM and proposed method increased as well. However, the SINR, PESQ and STOI of the CMB and LNR

showed an increase and decrease procedure. The reason for this is that the problem of the self-cancellation of the main lobe was aggravated with the increase in the iSNR. Moreover, the proposed method obtained better results for the SINR, PESQ and STOI, except for the IMB. This indicates that the proposed method effectively improved the performance of the CMB under different iSNRs. In addition, Figure 4d shows the results of the SDIs. We can find that as the iSNR increased, the SDIs of the CMB, HKB, LNR and BPR showed decreases. The reason for this is that the problem of the self-cancellation of the main lobe is aggravated with the increase in the iSNR. Unlike the SDIs of the rest of the methods, which were more stable, the IMB used the real DOA of the target source, the SMF, GLC, PFLI and NRP-TMMSE were degenerated into the fixed beamformer, and the IDL-SIM and the proposed method corrected the steering vector of the target source backward. In addition, the proposed method obtained a lower SDI except for the IMB. This indicates that the proposed method has a stronger robustness under different levels of noise.

### 4.5. Verification of the DOA Deduction through the Spatial Response Power

In this section, the effectiveness of the DOA deduction by the spatial response power is verified. Figure 5 shows the histograms and the errors of the DOA of the target source varying with two kinds of errors of the initial DOA of the target source, i.e., 5° and 10°. The other simulation settings were the same as in Section 4.2.

**Case 1:** Figure 5a shows the statistical histogram of the maximum value distribution of the spatial response power after the first iteration. We can find that the maximum value is located at 71.1°, rather than at 75° of the initial DOA of the target source. This confirms the effectiveness of the procedures of the backward correction of the steering vector of the target source and the DOA deduction through the spatial response power. Furthermore, Figure 5b shows that as the number of iterations increases, the error of the derived DOA of the target source gradually decreases; this indicates that the iteration of the DOA deduction can effectively reduce the error of the estimated DOA of the target source.

**Case 2:** Figure 5c shows that an error of 10° can also be effectively reduced to 2.1° after the first iteration. Similarly, Figure 5d also shows that as the number of iterations increases, the error of the derived DOA of the target source gradually decreases. This also indicates that the proposed method is applicable to the case that the DOA of the target source is seriously mismatched.

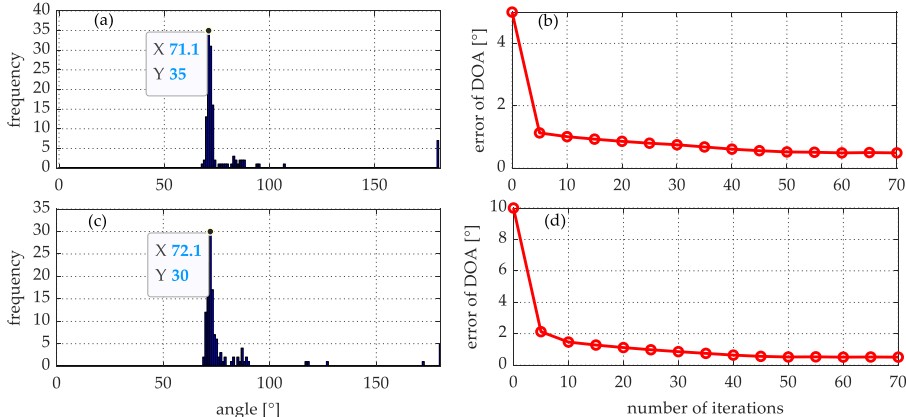

**Figure 5.** Histograms and DOA errors of the target source. (**a**) Histogram and (**b**) DOA error of the target source when the error of the initial DOA was 5°; (**c**) histogram and (**d**) DOA error of the target source when the error of the initial DOA was 10°.

### 4.6. Performance Analysis of the Optimization Modules

In order to analyze the performance of the three modules of the proposed method, we measured the performance of the IGLC-based MVDR beamforming method (labeled as IGLC), the Module 1-based MVDR beamforming method (labeled as Module 1), the Module

1 plus Module 2-based MVDR beamforming method (labeled as Modules 1 + 2), and the Module 1 plus Module 2 plus Module 3-based MVDR beamforming method (labeled as Modules 1 + 2 + 3). Figure 6 gives the evaluation results of the SINR, PESQ, STOI and SDI versus the iSNR.

The results of Figure 6 all show that the performance of the IGLC-based MVDR beamforming method is better than the IDL-SIM method, which confirms the effectiveness of the full loading-based IGLC method. Meanwhile, Figure 6 shows that the Module 1-based MVDR beamforming method is better than the IGLC-based MVDR beamforming method, which indicates that the backward correction of the steering vector of the target source based on the IGCL can effectively improve the performance of the CMB. Moreover, based on Module 1, when Module 2 was used for further optimization, the results of the SINR, PESQ, STOI and SDI improved, which confirms the effectiveness of the reconstruction of the covariance matrix based on steered response power. In addition, based on Module 1 and Module 2, when Module 3 was used, the results of the SINR, PESQ, STOI and SDI were further improved, which confirms the effectiveness of the derivation of the DOA of the target source based on the broadband spatial response power. Meanwhile, the gradual improvement results of Figure 6 also indicate the necessity of these three modules in the proposed method.

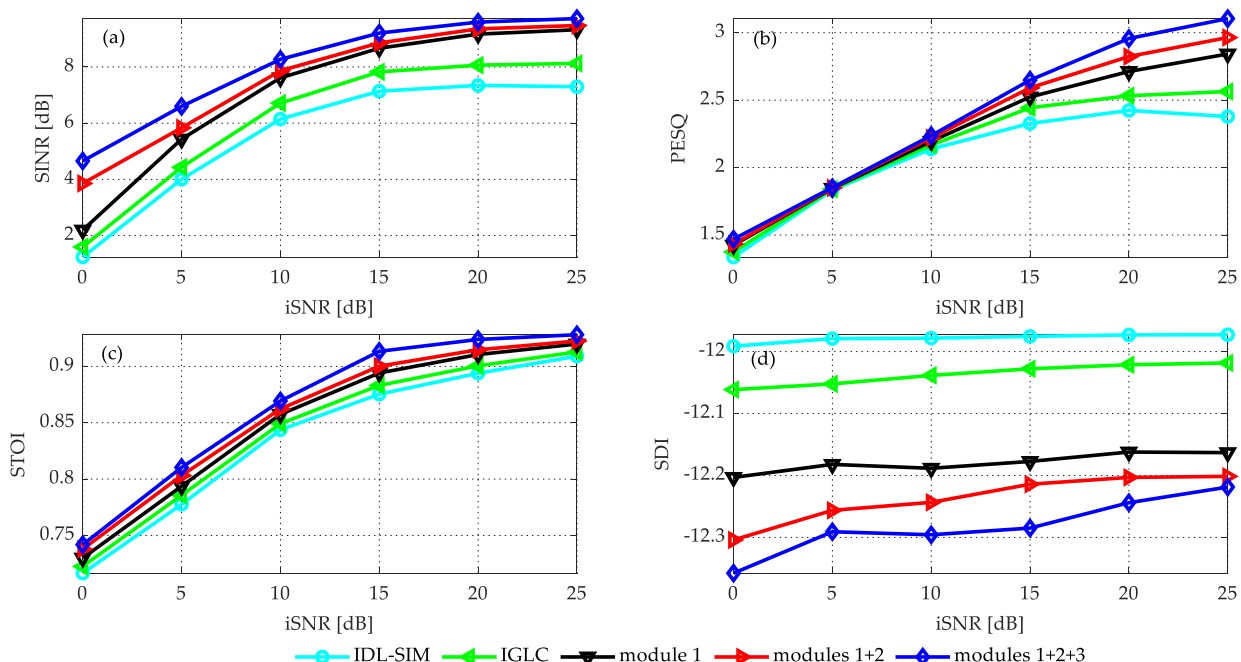

**Figure 6.** The evaluation results of the optimization modules under different iSNRs: (**a**) SINR; (**b**) PESQ; (**c**) STOI; (**d**) SDI.

## 5. Conclusions

This paper presented a full loading-based MVDR beamforming method by the backward correction of the steering vector of the target source and the reconstruction of the covariance matrix. In this method, based on the principle of diagonal loading, a full loading matrix was constructed, which improved the loaded covariance matrix approach to the CMIN by correcting the off-diagonal elements and eliminating the components of the target source. To reduce the mismatch between the steering vector and the target source, the weighting vectors of the CMB and the full loading-based MVDR beamformer were used to correct the steering vector of the target source backward. Furthermore, based on the uncertain set and the eigenvalue decomposition, a steered covariance matrix was built to further suppress the interference source. Moreover, spatial response power was used to derive a more accurate DOA of the target source which is helpful for obtaining a

more accurate steering vector of the target source and a more effective covariance matrix iteratively. We used a TIMIT corpus to verify the proposed method. The results showed that the proposed method effectively improved the performance of the CMB.

**Author Contributions:** Conceptualization, C.B. and J.Z.; methodology, C.B. and J.Z.; software, J.Z.; validation, C.B. and J.Z.; formal analysis, C.B. and J.Z.; investigation, J.Z.; resources, J.Z.; data curation, J.Z.; writing—original draft preparation, C.B. and J.Z.; writing—review and editing, C.B. and J.Z.; visualization, J.Z.; supervision, C.B. All authors have read and agreed to the published version of the manuscript.

**Funding:** This work was funded by the National Natural Science Foundation of China (Grant No. 61831019).

**Institutional Review Board Statement:** Not applicable.

**Informed Consent Statement:** Not applicable.

**Data Availability Statement:** Not applicable.

**Acknowledgments:** The authors are grateful to the thorough reviewers.

**Conflicts of Interest:** The authors declare no conflict of interest.

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
