# Peer review of "A Full Loading-Based MVDR Beamforming Method by Backward Correction of the Steering Vector and Reconstruction of the Covariance Matrix"

_applsci, doi:10.3390/app13010285_

Round 1
Reviewer 1 Report
- The paper is extremely difficult to read. Too many acronyms, not properly explained and motivated.
- The analyzed problem should be better clarified and explained in the Introduction,by a proper example, highlighting in a clear way The statistical targets;
- The many alternative methods cited are mostly not explained, even in intuitive terms;
- In Section 4.5, no definition of the reported performance metrics is provided.
The positive sides lie in the proposed procedure, which seems deep and articulated, although difficult to completely understand for the reader, and the simulation study, whose experiments on the Direction of Arrival (DOA) seems credible.
On my opinion, a hard reshaping work of language, presentation and motivation is needed, to properly understand the content and get the reader involved. I am not sure that ALL the targets of the paper are completely shown to have an optimal performance.
Reviewer 2 Report
The manuscript entitled "A Full Loading Based MVDR Beamforming Method by Back-ward Correction of the Steering Vector and Reconstruction of the Covariance Matrix" is worth to be published in Applied Sciences.
The DL based MVDR method sounds interesting for the readers.
The use of spatial response power to derive DOA is interesting.
Verification of the proposed method with TIMIT is good.
In the end, I sum up my comments by recommending the publication of this paper to the journal
Reviewer 3 Report
The work is interesting because with the development of telecommunications technology, many audio messages have appeared. However, if earlier specialists with good diction were specially selected for this purpose, now the circle of such people has significantly expanded. Therefore, the quality of audio transmission has decreased. The method proposed by the authors of the article to improve the sound signal quality is based on classical optimization procedures (1) - (5) and their (6) - (9) and (10) - (14) and correction of the covariance matrix.
But I am confused by the abundance of abbreviated names that make it difficult for mathematicians to read this text. And then, when writing formulas, notations that are inconvenient for mathematicians are used. Finally, the authors of the article use several different methods to improve the signal quality. However, the final results are not clearly presented enough to assess which of the proposed methods makes the greatest contribution to the solution of the task set by the authors.
Round 2
Reviewer 1 Report
sensibility p.1 r.35?
"this category of methods has ..." p.2, r.54
The REMAINING parts, p.2, r.78
(6)(9)(11)(13)(17)(19): all brackets misprinted.
Please solve!
p.6, r.215, "determined" not "solved"
Reviewer 3 Report
The authors have done a lot of work to correct the comments made. It was especially important for me to indicate how the individual steps of the proposed algorithm improve the operation of the system in question. And this comparison was made in sufficient detail.
